# Metabolic Alterations Caused by Simultaneous Loss of HK2 and PKM2 Leads to Photoreceptor Dysfunction and Degeneration

**DOI:** 10.3390/cells12162043

**Published:** 2023-08-10

**Authors:** Eric Weh, Moloy Goswami, Sraboni Chaudhury, Roshini Fernando, Nick Miller, Heather Hager, Sarah Sheskey, Vikram Sharma, Thomas J. Wubben, Cagri G. Besirli

**Affiliations:** Department of Ophthalmology and Visual Sciences, University of Michigan, Ann Arbor, MI 48105, USA; goswamim@med.umich.edu (M.G.); sraboni@med.umich.edu (S.C.); roshinis@med.umich.edu (R.F.); millnick@med.umich.edu (N.M.); hlindner@med.umich.edu (H.H.); ssheskey@med.umich.edu (S.S.); visharma@med.umich.edu (V.S.); twubben@med.umich.edu (T.J.W.)

**Keywords:** photoreceptor, metabolism, neuroprotection, aerobic glycolysis, HK2, PKM2

## Abstract

HK2 and PKM2 are two main regulators of aerobic glycolysis. Photoreceptors (PRs) use aerobic glycolysis to produce the biomass necessary for the daily renewal of their outer segments. Previous work has shown that HK2 and PKM2 are important for the normal function and long-term survival of PRs but are dispensable for PR maturation, and their individual loss has opposing effects on PR survival during acute nutrient deprivation. We generated double conditional (dcKO) mice lacking HK2 and PKM2 expression in rod PRs. Western blotting, immunofluorescence, optical coherence tomography, and electroretinography were used to characterize the phenotype of dcKO animals. Targeted and stable isotope tracing metabolomics, qRT-PCR, and retinal oxygen consumption were performed. We show that dcKO animals displayed early shortening of PR inner/outer segments, followed by loss of PRs with aging, much more rapidly than either knockout alone without functional loss as measured by ERG. Significant alterations to central glucose metabolism were observed without any apparent changes to mitochondrial function, prior to PR degeneration. Finally, PR survival following experimental retinal detachment was unchanged in dcKO animals as compared to wild-type animals. These data suggest that HK2 and PKM2 have differing roles in promoting PR neuroprotection and identifying them has important implications for developing therapeutic options for combating PR loss during retinal disease.

## 1. Introduction

Photoreceptor (PR) cell death is the ultimate cause of vision loss in many retinal diseases [1,2,3]. Identifying novel neuroprotective strategies for preventing PR loss is an urgent unmet need and an area of active research [1,3,4,5,6]. Current treatment approaches, such as gene-replacement therapy for inherited retinal diseases or complement therapies for late-stage, nonexudative age-related macular degeneration, either do not address the need for a broad therapeutic approach to PR neuroprotection or have minimal clinical benefit [7,8,9]. We and others have turned to investigating gene agnostic approaches to PR neuroprotection to circumvent these challenges [10]. A common mechanism thought to underlie PR loss in many retinal diseases is an imbalance in nutrient delivery to PR cells, which results in a metabolic crisis, causing loss of PRs that further pushes outer retinal metabolism away from homeostasis, worsening disease [11,12,13,14].

PRs are some of the most metabolically active cells in the body, requiring vast quantities of ATP to maintain the dark current [15]. In addition to these ATP demands, PRs require lipids, amino acids, and nucleotides to maintain the constant renewal of their outer segments [16]. To support these metabolic requirements, PRs convert up to 80% of the glucose that enters the outer retina into lactate by a process called aerobic glycolysis, or the Warburg effect [17,18,19]. Aerobic glycolysis is typically used by rapidly proliferating cells, such as cancer cells and stem cells, and has been shown to help support the immense biosynthetic needs required for cell division, whereas PRs use this unique metabolic adaptation to maintain outer segment renewal [20]. In the retina, two of the main regulators of aerobic glycolysis, hexokinase 2 (HK2), and pyruvate kinase muscle isoform 2 (PKM2), are expressed predominantly in PRs and have been linked to promoting PR neuroprotection during nutrient deprivation [4,21,22]. 

Previous research has shown that PRs rely on pyruvate kinase activity to maintain long-term health and function [22,23,24]. Loss of PKM2 via shRNA-mediated knockdown causes significant outer segment shortening, which is rescued with PKM1 over-expression, suggesting that pyruvate kinase function is necessary to maintain outer segments [24]. Conversely, when PKM2 is conditionally deleted from rod PRs, they significantly upregulate the PKM1 isozyme and initiate transcriptional alterations in central glycolytic gene expression. PRs exclusively expressing PKM1 develop and mature normally but display age-related chronic degeneration [22,23]. Most importantly, PKM1-expressing PRs are more resistant to nutrient-deprivation-induced cell death after experimental retinal detachment. At the same time, enhancing total pyruvate kinase (PK) activity through pharmacological activation of PKM2 in wild-type (WT) animals recapitulates this neuroprotective effect. These data indicate that intact PKM2 activity is critical for the long-term maintenance of PR health and that further enhancing total PK activity in PRs boosts their survival during stress [4,22].

HK2 performs the first step in glycolysis, phosphorylating glucose to glucose-6-phosphate, but in addition to this glycolytic role, HK2 has also been shown to be critical in reducing apoptotic activation [25,26]. Similar to PKM2 loss, deletion of HK2 from PRs does not affect the development or maturation of these cells [21,27,28]. Conditional deletion of HK2 from PRs leads to HK1 becoming the primary isozyme expressed in order to support glycolysis. PRs do not significantly alter central glycolytic gene regulation when preferentially expressing HK1, and these cells do not display any functional deficits at young ages. However, PRs expressing HK1 instead of HK2 undergo age-related degeneration with a corresponding decline in visual function as assessed by ERG. Furthermore, loss of HK2 leads to increased PR death following retinal detachment, demonstrating that the anti-apoptotic effects of HK2 is critical for cell survival during acute nutrient deprivation [21].

An unresolved question is whether both HK2 and PKM2 are required for PR health and function under normal physiological conditions or if they work together during nutrient deprivation to promote PR survival. Here, we present work investigating the simultaneous loss of both HK2 and PKM2 from rod PRs. These data show that both regulators of aerobic glycolysis play key roles in maintaining PR inner and outer segments and cell bodies while also preserving normal visual function. Metabolomics revealed significant alterations to central glucose metabolism with both targeted and stable isotope tracing methodologies without any alterations to gene regulation of glycolysis or differences in mitochondrial function. Most importantly, we found that simultaneous deletion of both genes resulted in neither enhanced survival nor enhanced cell death following acute nutrient deprivation.

## 2. Materials and Methods

### 2.1. Animals

All animals were treated in accordance with the Association for Research in Vision and Ophthalmology (ARVO) Statement for the Use of Animals in Ophthalmic and Vision Research. The protocol was approved by the University Committee on Use and Care of Animals of the University of Michigan (Protocol number: PRO00011135). Animals were housed in standard conditions under cyclic lighting conditions (12/12 h light/dark). Mice carrying a *Pkm* allele with exon 10 flanked by Lox-P sites were purchased from Jackson labs (*Pkm2^fl^*^/*fl*^, Jackson Laboratory, Bar Harbor, ME, USA; Strain# 024048). Mice carrying a floxed *Hk2* allele (*Hk2^fl/fl^*) were a gift from Dr. Mohanish Deshmukh and Dr. Timothy Gershon. These mice were bred to create animals homozygous for both *Hk2* and *Pkm* floxed alleles. The resulting animals were then crossed to mice carrying Cre recombinase under the control of the rhodospin promoter to selectively delete both *Hk2* and *Pkm2* from rod PRs [29]. Mice were confirmed to not carry the *rd8* allele and were maintained on the C57BL/6 background. *Hk2^fl/fl^; Pkm2^fl/fl^; Rho-Cre^+^* (double conditional knockout, dcKO) and *Hk2^wt/wt^; Pkm2^wt/wt^; Rho-Cre^+^* (wild-type, WT) animals were used for all experiments. Retinal tissue was extracted from freshly euthanized animals using the cut-and-pick method, being careful to avoid collection of ciliary body or RPE tissues [30]. Tissue was either used immediately (BaroFuse, Flow Cytometry, Pyruvate Kinase activity assay), immediately frozen on dry ice (Western blotting, metabolomics), or immersed in RNAlater (Qiagen; Hilden, Germany, Cat# 76104) for qRT-PCR. Whole eyes were extracted from freshly euthanized animals before being immersed in 10% neutral buffered formalin for histological analysis.

### 2.2. Western Blotting

Isolated retinal tissue was homogenized using a sonicator set at 20% amplitude with 1 s on/off pulse for 10 s in RIPA lysis buffer (Thermo Fisher Scientific, Waltham, MA, USA, Cat# 89900) supplemented with protease (Halt™ Protease Inhibitor Cocktail, Thermo Fisher, Cat# 87786) and phosphatase (Halt™ Phosphatase Inhibitor Cocktail, Thermo Fisher, Cat# 78420) inhibitors. The homogenized samples were then centrifuged at 10,000× *g* for 10 min at 4 °C. The supernatant was transferred to a fresh tube. Protein estimation was performed using the Pierce™ BCA Protein Assay Kit (Thermo Fisher, Cat# 23225). A total of 15 µg of protein was diluted with 4X Laemmli buffer (Bio-Rad; Hercules, CA, USA, Cat# 1610747) supplemented with β-mercaptoethanol (Millipore-Sigma; St. Louis, MO, USA, Cat# M6250) and heated to 95 °C before being centrifuged at 10,000× *g* for 5 min and then loaded onto a 4–20% Mini-PROTEAN^®^ TGX™ Precast Protein Gel (Bio-Rad, Cat# 4561094). Samples were separated and transferred to PVDF membrane using the Trans-Blot^®^ Turbo™ Transfer System (25 V for 30 min) (Bio-Rad, Cat# 1704150). Membranes were then immersed in 5% non-fat dry milk in TBST (Tris-buffered Saline (Bio-Rad, Cat# 1706435) supplemented with 0.1% Tween-20 (Thermo Fisher, Cat# 28320)). Primary antibodies and dilutions are listed in Table 1. All primary antibodies were diluted in 5% Bovine Serum Albumin (BSA; (Millipore-Sigma, Cat# A9647)) and added to membranes overnight at 4 °C with gentle rocking. Membranes were then washed in TBST before adding an appropriate secondary antibody diluted in 5% dry milk for 1 h at room temperature. Antibodies were detected using SuperSignal™ West Dura/Femto Extended Duration Substrate (Thermo Fisher, Cat# 34075 and 34094) with an Azure c500 imaging system (Azure Biosystems; Dublin, CA, USA).

### 2.3. Histology

Whole fixed eyes were embedded in paraffin and sectioned to 6 µM thickness and de-paraffinized following standard procedures. For immunofluorescence, de-paraffinized sections were subjected to antigen retrieval following standard procedures. Sections were then blocked with appropriate blocking buffer (either 10% normal goat serum in 1% BSA, or 10% BSA, in PBS supplemented with 1% triton-x 100 (PBST)) for 1 h before washing with PBST. Primary antibody (Table 1) was then added, and sections were incubated overnight in a 4 °C humidified chamber. Sections were then washed with PBST before an appropriate secondary antibody was added (Table 1). Sections were incubated for 1 h at room temperature before washing with PBST and coverslipped using Prolong Gold Antifade with DAPI (Thermo Fisher, Cat# P36935).

### 2.4. qRT-PCR

RNA was extracted from isolated retina using the RNeasy Mini Kit (Qiagen, Cat# 74104) following the manufacturer’s supplied protocol. Purified RNA was quantified using a Nanodrop 1000 (Thermo Fisher), and 400 ng was used as template for a cDNA synthesis reaction using the RT^2^ First Strand Kit (Qiagen, Cat# 330401) following the manufacturer’s instructions. The resulting cDNA was used as a template for a qRT-PCR reaction using the RT^2^ Profiler PCR Array for Mouse Glucose Metabolism (Qiagen, Cat# 330231 PAMM-006ZA). For each sample, 102 uL of cDNA was pre-mixed with 650 uL of 2X SYBR Green master mix (Qiagen, Cat# 330503) and 548 uL of ddH2O. Each well received 10 uL of pre-mix following the manufacturer’ guidelines. A CFX384 thermocycler (Bio-Rad) was used to perform PCR following the supplied cycling conditions and to obtain fluorescent intensity measurements from each well. The 2^−ΔΔCt^ method was used to obtain relative transcription levels between samples. The geometric mean of the *Gapdh* and *Hsp90ab1* Ct values was used to normalize data. A cycle threshold cutoff of 35 cycles was used to determine presence of transcripts.

### 2.5. Oxygen Consumption Measurement

Oxygen consumption rate (OCR) was determined using a BaroFuse (Entox Biosciences; UW, Diabetes Center, Seattle). Commercial Krebs-Ringer Solution, HEPES-buffered (KRB; Thermo Fisher, Cat# J67795. K2), was modified for perifusion by the addition of 0.1 g/100 mL Fatty Acid Free Bovine Serum Albumin (Millipore-Sigma, Cat# A9647) and 5.5 mM glucose (Millipore-Sigma, Cat# G8270) to mimic physiological levels in the retina. Oxygen content of the perifusion media was detected via a polymerized Pt (II) Oxygen Meso tetra (pentafluorophenyl) porphine dye coated on the glass chambers of the BaroFuse, which contained each tissue sample. The phosphorescence of the dye was measured as KRB flows past the tissue sample; the lifetime of the phosphorescence signal decay is dependent on oxygen tension which allows for real-time quantitative measurement of oxygen consumption. The oxygen and CO_2_ concentration of KRB entering the perifusion chamber was maintained by saturating the solution in a 21% oxygen, 5% CO_2_ atmosphere continuously throughout the entire period of experimentation. The entire chamber was maintained at 36 °C. Retinas were dissected as described above in Hank’s Balanced salt solution (Cytiva, Marlborough, MA, USA, Cat# SH30031FS) supplemented with 0.1 g/100 mL BSA. Each chamber contained the entire retina from a single eye. Oligomycin-A (Cayman Chemical, Ann Arbor, MI, USA, Cat# 11342), FCCP (Trifluoromethoxy carbonylcyanide phenylhydrazone, Cayman Chemical, Cat# 15218), and KCN (Potassium Cyanide, Thermo Fisher, Cat# 012136) were injected at various time points using an injection port. The protocol measures the OCR of wild-type and dcKO retinas for the following times: baseline, 90 min; Oligomycin-A (20 µM), 60 min; FCCP (1 µM), 45 min; and KCN (3 mM), 60 min. The OCR reaction was terminated through potassium cyanide application. A chamber with no tissue was used as the negative control for oxygen consumption.

### 2.6. PK (Pyruvate Kinase) Activity

An enzyme-coupled assay that measures absorbance at 340 nm to quantify depletion of NADH by lactate dehydrogenase (LDH) was used to determine pyruvate kinase (PK) activity [4,22]. Retina was harvested from animals following anesthetization and decapitation as described above. Tissue was lysed in RIPA Lysis and Extraction Buffer supplemented with protease inhibitors. Each assay was performed using 4 to 8 µL of retinal extract. Retinal extract was added to a solution containing 0.5 mM PEP, 1 mM ADP, 0.2 mM NADH, and 8 U of LDH in a buffer consisting of 50 mM Tris-HCl (pH 7.4), 100 mM KCl, and 5 mM MgCl_2_ as previously described [4,22,31]. A SPECTROstar Omega plate reader (BMG LABTECH Inc., Cary, NC, USA) was used to measure absorbance at 340 nm. The included MARS software v3.31 suite was utilized to determine initial reaction velocities. Data were normalized to the protein concentration of each sample and the relative activity was determined compared to WT eyes as previously described [4].

### 2.7. OCT and ERG

Animals were anesthetized using a mixture of ketamine and xylazine (90/10 mg/kg, ketamine/xylazine), and their pupils were dilated using 1% Tropicamide and 2.5% phenylephrine ophthalmic drops. For retinal thickness measurements, animals underwent optical coherence tomography (OCT) imaging using an Envisu-R SD-OCT imager (Leica Microsystems Inc., Buffalo Grove, IL, USA) as previously described [32]. Briefly, a 1.5 mm B-scan (1000 A-scans × 100 frames) and a 1.5 mm × 1.5 mm rectangular volume (1000 A-scans × 36 B-scans × 6 frames) were obtained. Frames were registered and averaged using the built-in software, and average retinal thickness was determined using the Diver 1.0 software with a 9 × 9 array. For ERG analysis, animals were dark-adapted overnight in a 24 h darkroom and then prepared as described above. ERG was assessed using a Diagnosys Celeris ERG instrument (Diagnosys LLC, Lowell, MA, USA) measuring both scotopic and photopic responses as previously described [22]. 

### 2.8. Metabolomics

For targeted metabolomics, mouse retina was harvested from 2-month-old animals and processed for metabolite extraction as previously described [33]. Briefly, the retina from both eyes of a single animal were extracted, rinsed in PBS, and snap-frozen on dry ice before processing. Metabolites were extracted from frozen tissue by homogenization in 80% methanol at −80 °C. Input was calculated based off total protein content and lyophilized by SpeedVac. Samples were analyzed by targeted LC-MS/MS via dynamic multiple reaction monitoring. Following mass determination, data were pre-processed with Agilient MassHunter Workstation Quantitative Analysis Software (B0900). These data were then further post-processed for quality control. Samples were normalized by the total intensity of all metabolites, then each metabolite abundance was normalized by the median of WT abundance levels for comparison, statistical analysis, and visualization. Statistical significance was determined using a two-tailed *t*-test with a threshold set to 0.05 [34,35].

For ^13^C-glucose labeling experiments, 2-month-old animals were injected with 2 g/kg uniformly labeled ^13^C-glucose intraperitoneally and retinas were harvested 45 min later and processed as described above [36]. m1, m2, m3, etc., refer to the extra mass conferred by ^13^C derived from the universally labeled glucose. A m1 molecule would have one ^13^C carbon, a m2 molecule would have 2 ^13^C carbons incorporated, a m3 molecule would have 3 ^13^C carbons incorporated, etc. The enzymatic breakdown pathways for glucose are well characterized, so the expected pattern of heavy carbons within each molecule is known when starting with a labeled molecule.

### 2.9. Retinal Detachment and TUNEL Flow Cytometry

Experimental retinal detachment was created as previously reported [21,22]. Briefly, mice were anesthetized with a mixture of ketamine/xylazine, and their eyes were dilated as described above. A drop of methylcellulose was applied to the cornea and a 3 mm glass coverslip was placed to flatten the cornea and allow visualization of the posterior chamber. A 25-gauge (G) microvitreoretinal knife was used to create a sclerotomy just posterior to the limbus. A 35 G beveled cannula attached to a nanofil syringe (World Precision Instruments, Sarasota, FL, USA Cat#s NF35BV-2, NANOFIL-100) were passed through the sclerotomy and a retinotomy was made to introduce the tip of the cannula into the subretinal space. Approximately 4 µL of Healon (Abbott Medical Optics, Santa Ana, CA, USA, Cat# 05047450842) was injected, which caused approximately one-half to two-thirds of the neural retinal to become detached from the underlying retinal pigment epithelium. Three days later, animals were euthanized, and their retinas were extracted as described above before processing for TUNEL staining and flow cytometry.

Retinas were processed for TUNEL staining and flow cytometry as previously described [37,38]. Retinas were digested into a single-cell suspension using a 0.25% Trypsin-EDTA solution (Thermo Fisher, Cat# 25200) supplemented with 400 µg of deoxyribose nuclease II (Worthington Biochemical, Lakewood, NJ, USA, Cat# LS002425). The resulting cell suspension was fixed using 4% paraformaldehyde for 20 min at room temperature before being permeabilized with 70% ice-cold ethanol at −20 °C overnight. Cells were rinsed and stained using the DeadEnd Fluorometric TUNEL assay (Promega, Madison, WI, USA, Cat# G3250) following the manufacturer’s instructions. As a negative control, retinal cell suspension from an attached retina was treated with staining solution without the rTDT enzyme. Stained cells were analyzed using an Attune Nxt flow cytometer with a minimum of 200,000 events detected for each sample on the FITC channel. Cells were initially gated by forward and side scatter to exclude events that may be representative of clumped cells or debris. The gates for analysis were set using the negative control at or below 1%. Data were analyzed using FCS Express v7.18.0021 (De Novo Software, Ontario, CA, USA).

## 3. Results

### 3.1. Successful Simultaneous Deletion of Both HK2 and PKM2 from Rod Photoreceptors

Previous work showed that deleting either HK2 or PKM2 in rod PRs did not significantly affect PR development or maturation [21,22,23,27,28]. To determine the effect of simultaneous loss of both HK2 and PKM2 on PR health and function, mice homozygous for both *Hk2* and *Pkm2* floxed alleles as well as Cre recombinase under the control of the rod-specific Rho promoter (dcKO mice) were generated. Western blotting on proteins from 3-month-old animals determined that both PKM2 and HK2 were nearly completely lost from the retina in dcKO animals (Figure 1A,B, Appendix A) with compensatory upregulation of HK1. PKM1 showed a trend in increased expression that did not reach statistical significance via Western blotting. Immunofluorescent staining showed specific loss of both HK2 and PKM2 from PR inner segments with a noticeable increase in signal for both HK1 and PKM1 in the inner segments of PRs only (Figure 1C–J). Since PKM1 is expressed highly throughout the inner retina, Western blotting may not be sensitive enough to detect an increase in PKM1 levels in dcKO PRs from total retina samples. These data are consistent with the data previously published for animals with single knockout of either gene [21,22,23,27]. Normal expression of HK2 and PKM2 in cone PRs was confirmed via co-staining with cone opsin in dcKO animals (OPN1MW, Figure 1D′,H′).

### 3.2. HK2 and PKM2 Are Required to Maintain PR Health and Function

As observed in Figure 1, PRs are not reliant on the simultaneous expression of HK2 and PKM2 for maturation after rhodopsin expression begins. To determine if these proteins are required for PR survival as animals age, dcKO animals were followed using optical coherence tomography (OCT). We found that by 3 months of age, both the outer nuclear layer and the inner segment/outer segment layers began to thin significantly in dcKO animals compared to WT animals (Figure 2A,B). This thinning progressed as animals aged and by 5 months of age, the total retinal thickness had decreased significantly. 

The early thinning of the inner/outer segments suggests that PR function may be affected prior to cell death. The outer segments of PRs are responsible for mediating the phototransduction cascade, which eventually leads to the sensation of vision. With the observation that the inner/outer segments of PRs in dcKO animals begin to shorten by 3 months of age, retinal function was assessed via electroretinogram (ERG) at 2 months (prior to the onset of degeneration) and 5 months of age (subsequent to the onset of degeneration). ERG recorded in both scotopic (Figure 3A,B) and photopic (Figure 3C–F) conditions showed alterations in PR function. Under scotopic conditions, no difference in a-wave or b-wave was observed at 2 months of age; interestingly, though, a significant increase in the scotopic b-wave of 5-month-old dcKO animals was observed (Figure 3A,B). Similarly, under photopic conditions, ERG did not show any statistically significant difference between WT and dcKO animals at 2 months of age. At 5 months of age, however, dcKO animals showed a significantly enhanced photopic a-wave and b-wave response (Figure 3C,D), as well as an enhanced response from a flickering stimulus (Figure 3E,F). We also compared WT animals at 2 and 5 months as well as dcKO animals at 2 and 5 months. As has been previously reported, we found a significant decrease in scotopic and photopic amplitudes for WT animals as they age [39]. When comparing 2-month-old to 5-month-old dcKO animals we did not identify any significant changes. The PR and IS/OS degeneration noted on OCT would be expected to reduce the ERG response. The paradoxical increase in ERG measurements, even in the presence of outer retinal degeneration, suggests that HK2 and PKM2 are important for maintaining retinal function within a physiologic range. 

### 3.3. Central Glucose Metabolism Is Significantly Altered in dcKO Animals

The data presented thus far indicate that HK2 and PKM2 are important for preserving normal PR survival and function. Previous work has also shown that single gene knockout can disrupt normal PR glycolytic gene regulation and mitochondrial function [21,22,23,27]; however, the importance of simultaneous deletion of both on PR glycolytic gene regulation and mitochondrial function is unclear. Initially, 84 genes involved in central glucose metabolism were examined using qRT-PCR (Appendix A). qRT-PCR demonstrated that 16 genes were significantly downregulated; however, their fold change was relatively low. It is therefore unclear if there is any functional significance to the observed gene-expression changes. In accordance with findings from single deletion of PKM2 [22], a significant increase in specific PK activity was found in dcKO animals. Yet, no changes to total lactate or ATP content were identified in 2-month-old dcKO animals (Figure 4A). In order to better identify differences in the metabolic state of the retina between WT and dcKO animals, unlabeled targeted metabolomics of 232 metabolites from major metabolic pathways (Appendix A) was performed using retina harvested from 2-month-old animals. A total of 85 metabolites from glycolysis, the pentose phosphate pathway (PPP), the tricarboxylic acid (TCA) cycle, and amino acid metabolism were quantified (Figure 4B).

A significant reduction in metabolites within central glycolysis (G6P: glucose-6-phosphate; DHAP: dihydroxyacetone phosphate), the TCA cycle (aconitate, α-ketoglutarate, succinate), and PPP (S7P: sedoheptulose 7-phosphate) was identified. Interestingly, a significant increase in both threonine and serine was noted in dcKO animals compared to WT animals (Figure 4C). Finally, a significant decrease in nicotinamide adenine dinucleotide (NAD^+^) was identified in dcKO animals (Figure 4D), which is an important metabolite in numerous pathways in central glucose metabolism. 

To further interrogate the metabolic changes identified above, isotope tracing metabolomics with uniformly labeled ^13^C-glucose was performed. We uncovered enrichment of glycolytic intermediates (Figure 5A) in retinas from dcKO animals. No detectable changes were seen in amino acid or TCA cycle metabolites (Figure 5B,C) between dcKO and WT animals. The decrease in the metabolite pool size of the glycolytic intermediates observed in the targeted experiments above reflects either decreased production or increased consumption in the dcKO mice. These tracing data suggest that the pool sizes are unlikely due to decreased production as the PRs of the dcKO mice demonstrate increased glucose utilization in this pathway.

### 3.4. Metabolic Changes Do Not Alter Mitochondrial Function in dcKO Animals

Considering the significant changes observed in central glucose metabolism and NAD^+^ abundance, mitochondrial function as determined by retinal oxygen consumption and expression levels of key mitochondrial proteins were assayed. The oxygen consumption rate (OCR) of total retina was measured using the BaroFuse (Figure 6A,B), which showed no significant alterations to fractional oxygen consumption in response to oligomycin (mitochondrial respiration) or FCCP (maximal respiratory capacity). The retinal expression levels of proteins in the electron transport chain (ETC) were also unchanged in dcKO animals (Figure 6C,D; Appendix A). As previously shown, PRs normally function near their maximal respiratory capacity [11,40], and switching to HK1 and PKM1 appears to have no effect on mitochondrial respiration or protein complement. 

### 3.5. Loss of HK2 and PKM2 Does Not Significantly Affect Cell Death Following Retinal Detachment

Increasing PK activity in PRs genetically via PKM2-to-PKM1 isoform switch or selective pharmacologic activation of PKM2 using a small molecule leads to PR neuroprotection following experimental retinal detachment [4,22]. In contrast, selective deletion of HK2 results in enhanced PR death after retinal detachment despite a compensatory increase in HK1 expression [21]. To determine if HK2/PKM2 neuroprotection pathways interact, we induced retinal detachment in 2-month-old dcKO and WT animals and collected whole retina at 3 days for TUNEL analysis via flow cytometry. The data shown in Figure 7 demonstrate that there is no significant difference in the percentage of TUNEL positive cells between WT and dcKO animals.

## 4. Discussion

Understanding the metabolic pathways that support retinal cell function and survival and how these pathways are rewired in retinal degenerative diseases is expected to provide a framework for developing novel and gene agnostic therapeutic paradigms to prevent vision loss. Previous work looking at single knockout of either HK2 or PKM2 in PRs suggests that altering expression or function of metabolic enzymes may be useful for preserving PRs during the metabolic stress experienced in retinal disease [4,21,22]. To determine if these pathways interact with each other, animals were generated with conditional deletion of both HK2 and PKM2 from rod PRs. The data presented here show that loss of these two enzymes results in compensatory upregulation of HK1 and PKM1. PRs develop normally in dcKO animals but display an age-dependent retinal degeneration phenotype that progresses at a faster rate than the degeneration seen in either single knockout alone. This degeneration is associated with significant alterations to glucose metabolism, as well as reduced total NAD^+^ content. Paradoxically, we found an enhanced ERG response in dcKO animals. Unlike single knockout models, simultaneous deletion of HK2 and PKM2 does not alter PR survival when faced with acute nutrient stress following experimental retinal detachment.

Previous data have shown that single deletion of either HK2 or PKM2 is dispensable for normal PR development and maturation but results in a progressive, age-related PR degeneration. [21,22,23,27,28]. Similarly, the data presented here show that co-expression of HK2 and PKM2 is not required for PR development and maturation but is necessary for maintaining early PR integrity and function. We found that dcKO animals showed thinning of the IS/OS layer by 3 months of age, whereas previous reports of single knockout animals did not report any IS/OS thinning until much later (10 months for HK2 [21] and 5 months for PKM2 [23] single knockout animals). The data presented here show that IS/OS biogenesis is further compromised in the absence of both enzymes and that the ONL thickness decreases much more rapidly than in single knockout animals. Compared to other models of retinal degeneration (e.g., *rd1*, *rd10*) [41], the PR degeneration in dcKO animals is slower, suggesting that HK1 and PKM1 can partially compensate for the loss of HK2 and PKM2. However, the accelerated degeneration in dcKO animals compared to their single KO counterparts indicates that each of the isoforms have hitherto unidentified but distinct functions that cannot be compensated. 

To better understand the role of these enzymes in PR function, we examined animals at 2 months of age to rule out any potential confounding variables due to PR degeneration. By looking at the PRs prior to degeneration, we can obtain a better insight into the changes leading to eventual photoreceptor loss due to HK2 and PKM2 loss. If we were to examine animals at a more advanced age, during degeneration, it is likely that more apparent differences would be identified; however, at least some of the changes would likely be due to PR death and not solely due to the loss of HK2 and PKM2. Since the Cre-mediated deletion of HK2 and PKM2 occurs very early (approximately post-natal day 12) [29], the metabolic changes are expected to be present at the time of PR maturation, prior to the onset of degeneration. 

Removing key regulatory enzymes of aerobic glycolysis that have been previously demonstrated to support aerobic glycolysis and biosynthesis clearly affects normal photoreceptor survival and function [21,22,23,24,27,28]. The metabolic alterations observed in dcKO animals likely play a key role in accelerated PR degeneration compared to single knockout animals. HK2 and PKM2 are key regulators of aerobic glycolysis [42,43]. PKM2 can be regulated to have either high or low activity, whereas PKM1 only functions in a high-activity state [44]. If PRs cannot regulate PK activity, glycolytic intermediates may not be able to accumulate and be subsequently shunted into biosynthesis [20]. The targeted metabolomics data did show a decrease in upstream glycolytic intermediates, but this is not due to lack of glucose utilization, as demonstrated by stable-isotope-labeled metabolomics, which may signal increased glucose consumption. Curiously, a reduction in TCA cycle intermediates was also observed. The TCA cycle integrates glucose, amino acid, and lipid metabolism, and PRs have been demonstrated to use fuels beyond glucose to support the TCA cycle [12,45]. Hence, it is possible that metabolic reprogramming diverts TCA cycle intermediates back into PEP to support anabolic metabolism in an attempt to support OS biogenesis, considering that the key glycolytic node PKM2 has been lost. A similar phenomenon has been observed in cancer cell metabolism, upregulating PEPCK expression to divert TCA cycle metabolites back in glycolysis for biomass synthesis [46].

^13^C-glucose-labeled metabolomics showed an increase in the labeled fraction of metabolites in glycolysis only, confirming that glucose is being preferentially diverted to lactate production in dcKO animals instead of biomass production. Conversely, when we measured the mitochondrial function of whole retina, we did not see any changes to OCR with either oligomycin or FCCP administration compared with WT retinas. These data suggest that rewiring metabolism towards oxidative phosphorylation cannot further enhance PR mitochondrial function as PRs operate at or near their maximal OCR at baseline [11,40]. This would suggest that if dcKO PRs are increasing the incorporation of glucose carbons into glycolytic intermediates, any extra would have to be exported as lactate as there is little, if any, reserve capacity to increase oxidative phosphorylation to use the excess carbon from glucose. As we did not see an increase in total lactate levels from dcKO retinas from either total lactate quantitation or by metabolomics analysis, we speculate that the increased intracellular lactate in PRs of dcKO is rapidly exported into the extracellular space and metabolized by the RPE, thus preventing the detection of any potential change in lactate levels via retinal metabolomics [47,48]. qRT-PCR analysis found significant downregulation of several genes associated with central glycolysis. However, as the fold changes are relatively small (<1.5-fold down), it is not entirely clear if these gene transcription changes are contributing to the metabolic rewiring, even in the presence of the metabolomics data showing significant alterations to cellular metabolism. Importantly, a significant decrease in NAD^+^ was found in dcKO retinas. This could indicate decreased production or increased utilization of NAD^+^, which is an important co-factor in numerous steps of cellular metabolism [49]. The data presented above suggest increased consumption due to increased glycolytic flux and diversion of TCA cycle metabolites for anaplerosis. Alterations to NAD^+^ homeostasis have been linked to AMD progression [50]. Our data suggest that major glycolytic rewiring and reduced NAD^+^ are potential causes of the age-related PR loss in the absence of HK2 and PKM2. 

Despite the early thinning of the IS/OS layer, ERG amplitudes were paradoxically increased in dcKO mice in response to flash stimuli. dcKO animals displayed an enhanced scotopic b-wave response at the mixed cone-rod threshold stimulus. They also displayed enhanced a- and b-wave photopic responses, along with an enhanced flicker response. It has been reported that changes to nutrient availability can alter the ERG response in rodents, cats, and humans [51,52,53]. A potential explanation for this paradoxical ERG response in the presence of IS/OS thinning is that increased levels of HK1 and PKM1 in PRs allow these cells to use the available glucose more readily, thereby mimicking the enhanced ERG response associated with increased blood glucose levels. Our ^13^C-labeled glucose metabolomics demonstrate that the dcKO animals have increased utilization of glucose in glycolysis as compared to WT animals. It has also been shown that intravitreal delivery of glucose can delay the loss of ERG response postmortem in rats and preserve the ERG for longer than glutamine or lactate alone. These data show that glutamine and lactate are not preferred fuel sources for retina in vivo to maintain the ERG response and suggest that diverting glucose through glycolysis, and not the TCA cycle, is beneficial for the ERG response [54]. At the same time, the b-wave amplitude was significantly increased despite inner retinal neurons not being affected by the genetic knockout. Similarly, photopic ERG was enhanced, despite cone PRs maintaining normal HK2 and PKM2 expression. Data obtained from human patients with cone-only retinal degeneration (e.g., achromatopsia) have shown defects in rod-mediated ERG, despite rods not being affected by the genetic mutation, and is not explained by a decrease in rod photoreceptor numbers [55]. As it has been shown that changes in cone function can influence rod function, it is plausible that the opposite may also be true. Indeed, patients with rod-driven retinal degeneration (e.g., retinitis pigmentosa) have been shown to display changes to cone function prior to degeneration [56]. The metabolic microenvironment within the retina, between cones, rods, and RPE cells, is likely altered due to these disease processes, which may be responsible for the non-cell autonomous changes in function observed.

Detachment of the neural retina from the underlying RPE induces an acute nutrient deprivation in PRs, with decreased glucose availability as seen by alterations in glycolytic and mitochondrial protein levels [57]. The TUNEL data shown here indicate that PRs expressing both HK1 and PKM1 show the same susceptibility to cell death compared to wild-type PRs expressing HK2 and PKM2. HK2 is known to have a moonlighting function as an anti-apoptotic protein, preventing the opening of the mitochondrial transition pore [25]. HK1 has been shown to have similar anti-apoptotic activity in non-retinal cells, but an increased expression of HK1 following the deletion of HK2 was unable to compensate for HK2-mediated neuroprotection, leading to increased PR death following retinal detachment [58,59,60]. In contrast, deletion of PKM2 from PRs results in expression of the PKM1 isoform, leading to enhanced total PK activity and a neuroprotective effect after experimental retinal detachment that can be recapitulated with pharmacological activation of PKM2. These data suggest that enhanced total PK activity likely underlies the improved PR survival in those animals. If the neuroprotection mediated by enhanced PK activity functions through the anti-apoptotic effects of HK2, enhanced cell death would be expected due to loss of control over apoptotic progression. Alternatively, if HK2 and enhanced PK activity both act on the same pathway, the enhanced PK activity afforded by PKM1 may be able to prevent the enhanced death when HK2 is replaced by HK1 [4,22]. Interestingly, both HK2 and PKM2 have been shown to participate in the mTOR pathway. HK2 can sequester mTORC1 to activate autophagy [61], which HK1 cannot do. PKM2 is a downstream target of mTORC1 and promotes mTORC1 activation via a positive feedback loop [62]. Hyperactive mTORC1 is protective in models of inherited retinal degeneration but causes AMD-like phenotypes in wild-type animals [63,64]. Further investigation is needed to determine if HK2 and PKM2 have competing roles in regulating mTORC1 to promote cell growth or autophagy and how those processes regulate PR health during nutrient stress.

## 5. Conclusions

In conclusion, simultaneous knockout of HK2 and PKM2 in PRs leads to compensatory increase in HK1 and PKM1 expression, early shortening of PR inner/outer segments, age-related PR degeneration, and paradoxical enhancement of scotopic and photopic ERG. Mutant PRs exclusively expressing HK1 and PKM1 showed significant alterations in central glucose metabolism without any detectable changes in mitochondrial function. Selective deletion of HK2 and PKM2 did not impact PR survival during acute nutrient deprivation secondary to experimental retinal detachment. It is clear that HK2 and PKM2 play distinct roles in promoting both PR cell function, as well as survival. There is obvious therapeutic potential in identifying these roles to slow or prevent the vision loss associated with degenerative retinal disease. By better understanding the non-enzymatic activities of these proteins, it may be possible to selectively enhance their neuroprotective functions without affecting their normal duties for PR maintenance.

## Figures and Tables

**Figure 1 cells-12-02043-f001:**
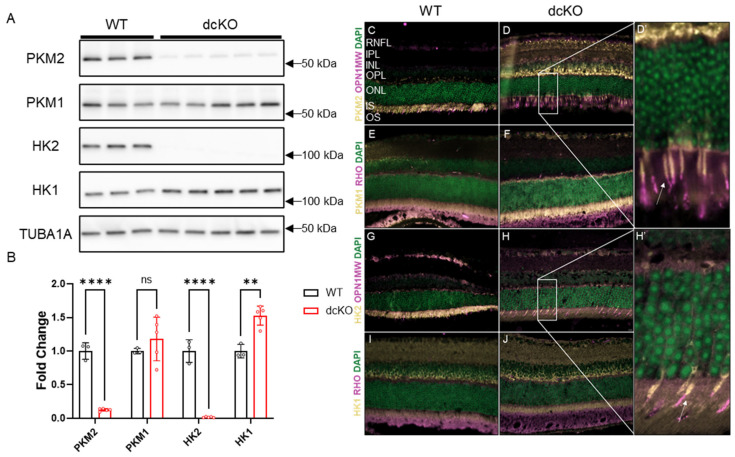
PKM2 and HK2 are deleted from rods only with upregulation of HK1 and PKM1. (**A**) Western blot showing significant reduction in PKM2 and HK2 proteins levels with significant upregulation of HK1, quantified in (**B**). *n* = 3–5, mean ± SEM. (**C**,**D**,**D′**) Staining showing loss of PKM2 from rod photoreceptors with expression remaining in cone inner segments. White arrow in (**D′**) indicates cone inner segment. (**E**,**F**) Immunofluorescence staining showing an increase in PKM1 expression in photoreceptor inner segments in dcKO animals. (**G**,**H**,**H′**) Staining showing loss of HK2 from rod photoreceptors with expression remaining in cone inner segments. White arrow in (**H′**) indicates cone inner segment. (**I**,**J**) Staining showing increase in HK1 expression in photoreceptor inner segments in dcKO animals. RHO staining indicates rod photoreceptor outer segments and OPN1MW staining indicates cone photoreceptors. ns, not significant; **, *p* ≤ 0.01; ****, *p* ≤ 0.00001.

**Figure 2 cells-12-02043-f002:**
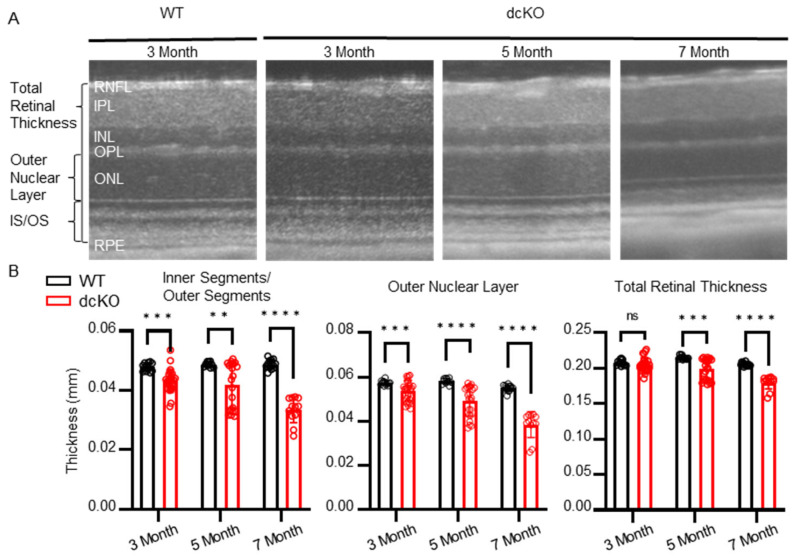
Loss of HK2/PKM2 causes early inner segment/outer segment and outer nuclear layer thinning. (**A**) Representative OCT b-scans and (**B**) quantitation showing decrease in total retinal thickness by 5 months of age. dcKO animals show decrease in outer nuclear layer and inner segment/outer segment thickness by 3 months of age. *n* = 10–26 per group, mean ± SEM. ns, not significant; **, *p* ≤ 0.01; ***, *p* ≤ 0.001; ****, *p* ≤ 0.00001.

**Figure 3 cells-12-02043-f003:**
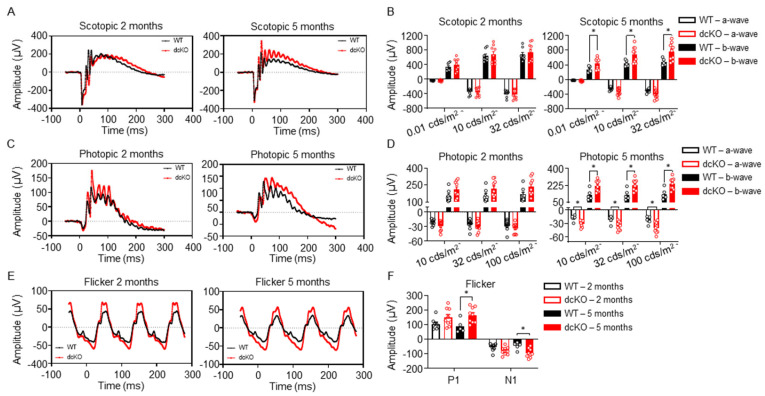
HK2 and PKM2 are required to maintain normal photoreceptor response to light stimulus. (**A**,**B**) Representative scotopic ERG (32 cds/m^2^) showing an increase in b-wave amplitude in dcKO animals by 5 months of age (left) and quantitation of ERG response (right). (**C**,**D**) Representative photopic ERG (1 Hz 100 cds/m^2^) showing an increase in both a- and b-wave amplitude in dcKO animals by 5 months of age (left) and quantitation of ERG response (right). (**E**,**F**) Representative flicker ERG (9.9 Hz 20 cds/m^2^) showing an increase in both N1 and P1 amplitude in dcKO animals by 5 months of age (left) and quantitation of ERG response (right). *n* = 8–9 per group, mean ± SEM. *—*p* ≤ 0.05.

**Figure 4 cells-12-02043-f004:**
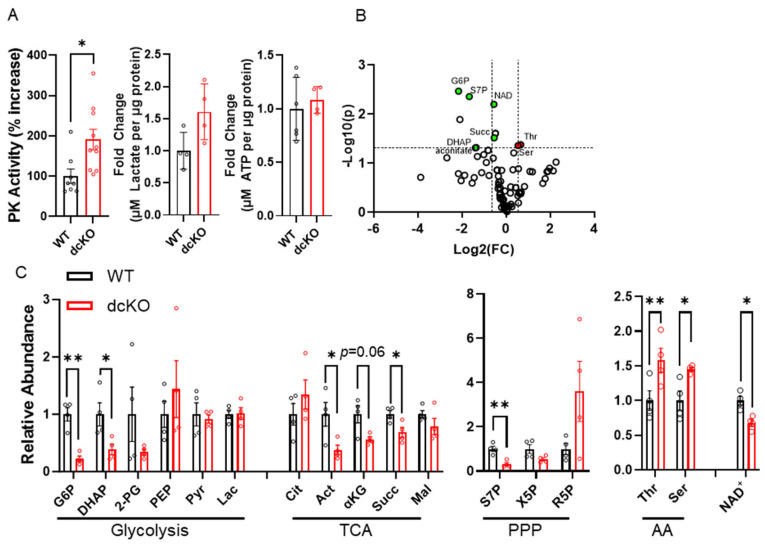
Loss of HK2 and PKM2 causes significant changes in central glucose metabolism at 2 months of age. (**A**) Total pyruvate kinase activity, total lactate content, and total ATP content measured from whole retina. *n* = 4–10 per group, mean ± SEM. (**B**) Volcano plot of 85 metabolites identified through targeted LC-MS/MS analysis. Horizontal dashed line indicates *p* ≤ 0.05. Vertical dashed lines indicate ± 1.5-fold change. *n* = 4 per group. (**C**) Graphs showing significantly altered metabolites identified via targeted metabolomics in respective pathways. *n* = 4 per group, mean ± SEM. AA: amino acids; PPP: pentose phosphate pathway; TCA: tricarboxylic acid cycle. G6P: glucose-6-phosphate; DHAP: dihydroxyacetone phosphate; 2-PG: 2-phosphoglycerate; PEP: phosphoenol pyruvate; Pyr: pyruvate; Lac: lactate; Cit: citrate; Act: aconitate; α-Kg: alpha-ketoglutarate; Succ: succinate; Mal: malate; S7P: sedoheptulose 7-phosphate; X5P: xylulose-5-phosphate; R5P: Ribose-5-phosphate; Thr: threonine; Set: serine; NAD: nicotinamide adenine dinucleotide. *—*p* ≤ 0.05, **—*p* ≤ 0.01.

**Figure 5 cells-12-02043-f005:**
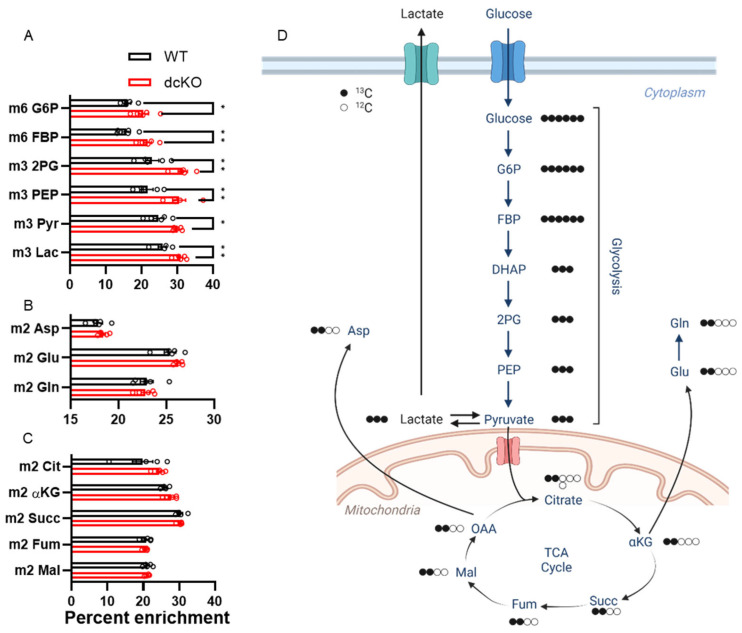
^13^C-labeled metabolomics show significant enrichment in glycolytic metabolites in the retina of dcKO mice. Graphs showing percent enrichment for metabolites within (**A**) glycolysis, (**B**) amino acids, and (**C**) TCA cycle. Only metabolites within glycolysis had significantly increased labeling from ^13^C-glucose. *n* = 5 per group, mean ± SEM. (**D**) Schematic pathway showing fate of ^13^C derived from universally labeled glucose. *, *p* ≤ 0.05; **, *p* ≤ 0.01. 2PG: 2-phosphoglycerate; αKG: alpha-Ketoglutarate; Asp: Aspartate, DHAP: Dihydroxyacetone phosphate; FBP: fructose 1,6-bisphosphate; Fum: fumarate; G6P: glucose-6-phosphate; Glu: glutamate; Gln: glutamine; Lac: lactate; Mal: malate; OAA: oxaloacetate; PEP: phosphoenolpyruvate; Pyr: pyruvate; Succ: succinate.

**Figure 6 cells-12-02043-f006:**
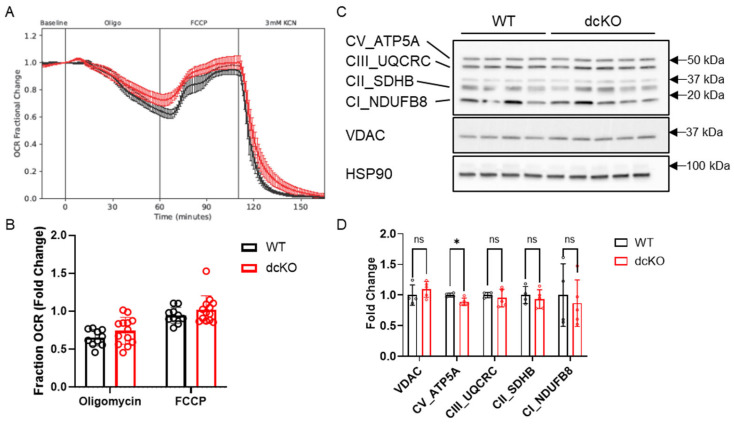
Loss of HK2 and PKM2 does not affect mitochondrial function. (**A**) Oxygen consumption rate and (**B**) quantitation showing no difference in fold change in oxygen consumption between 2-month-old dcKO and WT animals when exposed to oligomycin or FCCP. Data for WT or dcKO are normalized to their respective baseline values. *n* = 10–13 per group, mean ± SEM. (**C**) Western blot and (**D**) quantitation examining key mitochondrial proteins and oxidative phosphorylation components. *n* = 4–5 per group, mean ± SEM. ns, not significant; *, *p* ≤ 0.05.

**Figure 7 cells-12-02043-f007:**
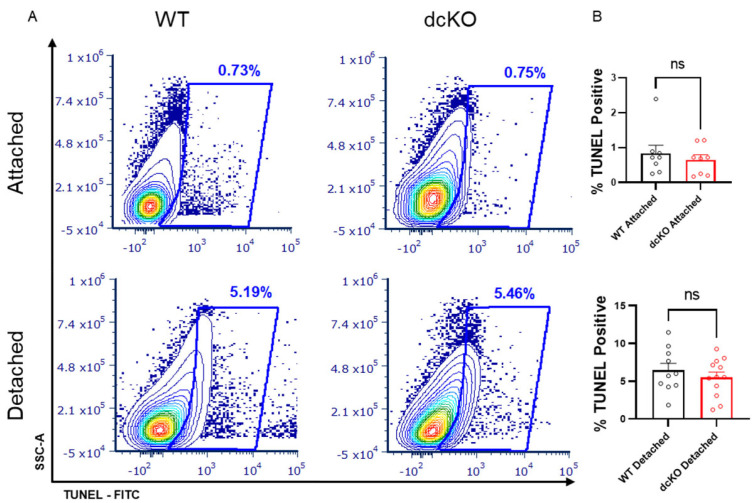
Loss of HK2 and PKM2 does not alter photoreceptor survival after experimental retinal detachment. (**A**) Flow cytometry contour plots of TUNEL-stained cells in either attached or detached retinas from dcKO and WT mice at 2 months of age. (**B**) Quantification of flow cytometry showing percentage of total events positive for TUNEL staining. Data show no difference in TUNEL-positive cells in attached retina or retina 3 days post-experimental retinal detachment between dcKO and WT mice at 2 months of age. *n* = 10–12 per group, mean ± SEM. ns, not significant.

**Table 1 cells-12-02043-t001:** List of antibodies used.

Target	Dilution	Supplier
HK1	1:1000 (WB), 1:2000 (IF)	CST (Cat #2024)
HK2	1:1000 (WB, IF)	CST (Cat #2867)
HSP90	1:1000 (WB)	CST (Cat #4877)
OPN1MW	1:400 (IF)	Santa Cruz (Cat #sc-22117)
PKM1	1:1000 (WB, IF)	CST (Cat #7067)
PKM2	1:1000 (WB, IF)	CST (Cat #4053)
RHO	1:1000 (IF)	Abcam (Cat #ab5417)
TUBA1A	1:2000 (WB)	Millipore Sigma (Cat #T6199)
VDAC	1:1000 (WB)	CST (Cat # 4866)
HRP anti-rabbit	1:2000 (WB)	Dako (Cat #P0447)
HRP anti-mouse	1:2000 (WB)	Dako (Cat #P0448)
Alexa594 anti-rabbit	1:1000 (IF)	Thermo Fisher (Cat #A11037)
Alexa488 anti-mouse	1:1000 (IF)	Thermo Fisher (Cat #A10680)
Total OXPHOS proteins	1:4000 (WB)	Abcam (Cat #ab110413)

CST: cell signaling technology; IF: immunofluorescence; WB: Western blot; HRP: horseradish peroxidase.

## Data Availability

All data will be made available upon request to the corresponding author.

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
