# Peer review of "Metabolic Alterations Caused by Simultaneous Loss of HK2 and PKM2 Leads to Photoreceptor Dysfunction and Degeneration"

_cells, 2023, doi:10.3390/cells12162043_

Round 1

Reviewer 1 Report

In this paper, the authors investigate wehther the simultaneous loss of both HK2 and PKM2 in rod photoreceptors will have am impact on PR survival and metabolism.

Despite some negative and contradictory results, the paper is well written and I would like to acknowledge the effort that the authors have put in the (long) discussion to explain these results.

I have some major comments:

1) My major concern is the use of 2 month old animals to perform most of the experiment in the paper. As shown in Figure 3, the retinal thickness is progressively decreased from 3 month to be more apparent at 7 month of age. However, the authors decided to use 2 month old animals to perform the metabolomics and all subsequent metabolic related experiments, despite that ERG at this age shows no difference (Figure 3). In my opinion, using older animals at 5 or 7 month of age would have more apparent differences, that were not apparent at 2 month. How the authors can explain this choice?

2) in paragraph 3.1., what is the age of animals used? Did the authors perform qPCR to detect PKM2, PKM1, HK2 and HK1 expression levels? Why PKM2/RHO or HK2/RHO not shown? in Figure 1D, PKM2 seems to be highly expressed in dcKO in other retinal layers as compared to the WT. Do you have any explanation for this observation as this also might affect the results observed. I also suggest to include retinal layers annotation on Figure 1.

3) Figure 3: Why the authors did not perform ERG on 7 month old animals?

Minor comments:

1) The title is too long, maybe you could consider to shorten it.

2) Line 39: worsening "the" disease.

3) Figure 4: "E" is misplaced, Line 335 Fig. 4C should be 4E

4) Line 419: ...IS/OS "thinning"... not "thing"

Author Response

We would like to thank the reviewers for their time and careful review of our article, previously titled: “HK2 and PKM2 are Critical for Photoreceptor Survival and Simultaneous Loss Negates Their Individual Neuroprotective Effects During Retinal Stress”. Please find below our point-by-point rebuttal to address the concerns raised by each reviewer.

Reviewer #1

In this paper, the authors investigate whether the simultaneous loss of both HK2 and PKM2 in rod photoreceptors will have an impact on photoreceptor survival and metabolism.

Despite some negative and contradictory results, the paper is well written and I would like to acknowledge the effort that the authors have put in the (long) discussion to explain these results.

We thank the reviewer for their time in critically analyzing our manuscript and providing insightful feedback.

I have some major comments:

1) My major concern is the use of 2 month old animals to perform most of the experiment in the paper. As shown in Figure 3, the retinal thickness is progressively decreased from 3 month to be more apparent at 7 month of age. However, the authors decided to use 2 month old animals to perform the metabolomics and all subsequent metabolic related experiments, despite that ERG at this age shows no difference (Figure 3). In my opinion, using older animals at 5 or 7 month of age would have more apparent differences, that were not apparent at 2 month. How the authors can explain this choice?

We thank the reviewer for the concern and thoughtfulness. The reason we examined animals at 2-months for most experiments in this paper was to rule-out any potential confounding variables due to photoreceptor degeneration. By looking at the photoreceptors prior to degeneration we can obtain a better insight into the changes leading to eventual photoreceptor loss due to HK2 and PKM2 loss. We agree that if we were to examine animals at a more advanced age, during degeneration, that we would likely identify more apparent differences; however, we believe it is likely that at least some of the changes will be due to photoreceptor death, and not solely due to the loss of HK2 and PKM2. Our goal was to better understand how HK2 and PKM2 contribute to normal photoreceptor maintenance, which we believe is best achieved by examining metabolism/oxygen consumption/etc prior to photoreceptor death. Since the Cre-mediated deletion of HK2 and PKM2 occurs very early (approximately post-natal day 12), the metabolic changes are expected to be present at the time of photoreceptor maturation, obviating the need for examining later time points, after the onset of degeneration.

2) in paragraph 3.1., what is the age of animals used? Did the authors perform qPCR to detect PKM2, PKM1, HK2 and HK1 expression levels? Why PKM2/RHO or HK2/RHO not shown? in Figure 1D, PKM2 seems to be highly expressed in dcKO in other retinal layers as compared to the WT. Do you have any explanation for this observation as this also might affect the results observed. I also suggest to include retinal layers annotation on Figure 1.

We apologize for the oversight. The samples tested for Figure 1 were from 3-month old WT and dcKO animals. This clarifying detail has been added to paragraph 3.1, line 268.

We did not perform qRT-PCR to detect changes in transcript levels for PKM2, PKM1, HK2, or HK1 as the Western blot data as well as immunofluorescence both clearly show reduction in protein levels for HK2 and PKM2, confirming rod photoreceptor knockout, as well as an increase in HK1 in rod photoreceptors. The non-significant change in PKM1 levels as measured by Western blotting is consistent with previous data published by our group for the single PKM2 knockout animal. The immunofluorescence data clearly shows an increase in PKM1 signal in the inner segments of photoreceptors, which is not present in WT animals. We believe that the reason our Western blotting data shows a non-significant increase in PKM1 protein is due to the fact that we are using whole retinal lysate, and therefore examining all cells of the retina and not just photoreceptors. PKM1 is expressed highly throughout the inner retina, and therefore the increase in PKM1 in photoreceptors following the deletion of PKM2 is diluted out when assaying total retinal lysate (PMID: 29259242). Adding qRT-PCR data will likely not add any additional information as was the case when we examined Hk2 transcript levels in the HK2 single knockout animals (no change in transcript level despite near total loss of HK2 protein, PMID: 32499533).

We do not show any data from the HK2 or PKM2 single knockout models, as these data have been previously published by our group (PMID: 29259242, 32499533). We reference these studies, as well as those from other groups who have also examined the single knockout animals, throughout the manuscript to help with the interpretation of the data presented in our manuscript.

We thank the reviewer for the suggestion and have added annotations for retinal layers to Figure 1 C-J. The apparent enhanced expression seen in the inner retina of PKM2 is due to increased exposure time to more easily distinguish the cone photoreceptors, which are maintaining expression of PKM2. The Western blot data confirms that PKM2 is not upregulated in the inner retina of these animals.

3) Figure 3: Why the authors did not perform ERG on 7 month old animals?

We thank the reviewer for their comment. Our primary goal with these studies was to measure the effects of HK2 and PKM2 loss on photoreceptor function early in the degeneration, prior to significant photoreceptor loss, as further degeneration in older animals would lead to decreased ERG due to the loss of photoreceptors, thus confounding our results.

Minor comments:

1) The title is too long, maybe you could consider to shorten it.

We have changed the title to: “Metabolic alterations caused by simultaneous loss of HK2 and PKM2 lead to photoreceptor dysfunction and degeneration”.

2) Line 39: worsening "the" disease.

The context of this sentence is referring to any degenerative photoreceptor disease, and therefore not specifying any single retinal condition in particular.

3) Figure 4: "E" is misplaced, Line 335 Fig. 4C should be 4E

We apologize for this typographical error. An updated Figure 4 removing the errant “E” has been included. There is no panel E for Figure 4.

4) Line 419: ...IS/OS "thinning"... not "thing"

We thank the reviewer for pointing out this typographical error which we have corrected.

In addition to the changes listed above, we have gone through the entire manuscript to enhance clarity without altering the conclusions of the original manuscript. We have also included an updated description for the retinal detachment and flow cytometric methods (lines 233-260) that were inadvertently omitted from our original submission. We would once again like to thank the reviewer for generously offering their time and services to critically analyze and provide helpful feedback to improve our manuscript.

Reviewer 2 Report

In this manuscript, the authors describe their characterization of mice with the rod photoreceptor-specific knockdown of two important regulators of aerobic glycolysis, Pkm2 and Hk2. This study expands on previous work from this research group as well as other groups on the importance of these proteins on rod photoreceptor health. Their findings that the genetic ablation of Pkm2 and Hk2 does not alter the survival of photoreceptors after experimental retinal detachment is intriguing, especially since PKM2 activation protects against apoptosis (PMID: 32076076) and HK2 is required for apoptosis in this model (PMID: 32499533). Furthermore, the increase of ERG responses in the Pkm2 and Hk2 double knockout mice when they exhibit a photoreceptor degeneration is also an interesting finding. However, the lack of rod-specific Pkm2 and Hk2 single knockout mice dramatically lessens the impact of this work. Instead, the authors rely on citing previous studies on the descriptions of these genotypes in this manuscript. The lack of these controls makes it difficult to interpret if there is an epistatic or additive relationship between PKM2 and HK2 in the retina.  

Author Response

We would like to thank the reviewers for their time and careful review of our article, previously titled: “HK2 and PKM2 are Critical for Photoreceptor Survival and Simultaneous Loss Negates Their Individual Neuroprotective Effects During Retinal Stress”. Please find below our point-by-point rebuttal to address the concerns raised by each reviewer.

Reviewer #2

In this manuscript, the authors describe their characterization of mice with the rod photoreceptor-specific knockdown of two important regulators of aerobic glycolysis, Pkm2 and Hk2. This study expands on previous work from this research group as well as other groups on the importance of these proteins on rod photoreceptor health. Their findings that the genetic ablation of Pkm2 and Hk2 does not alter the survival of photoreceptors after experimental retinal detachment is intriguing, especially since PKM2 activation protects against apoptosis (PMID: 32076076) and HK2 is required for apoptosis in this model (PMID: 32499533). Furthermore, the increase of ERG responses in the Pkm2 and Hk2 double knockout mice when they exhibit a photoreceptor degeneration is also an interesting finding. However, the lack of rod-specific Pkm2 and Hk2 single knockout mice dramatically lessens the impact of this work. Instead, the authors rely on citing previous studies on the descriptions of these genotypes in this manuscript. The lack of these controls makes it difficult to interpret if there is an epistatic or additive relationship between PKM2 and HK2 in the retina.  

We thank the reviewer for carefully reading our manuscript and for the positive comments on the phenotype we have reported. We agree with the reviewer that it is important to interpret our results in the context of the single knockout animals; however, as the single knockout HK2 and PKM2 phenotypes have been previously reported by our group and others, we believe that repeating previously published work would not significantly add to the work presented here. We reference those previously reported data throughout this manuscript to draw parallels and contrasts to interpret the results of the data presented herein.

Additionally, we have gone through the entire manuscript to enhance clarity without altering the conclusions of the original manuscript. We have also included an updated description for the retinal detachment and flow cytometric methods (lines 233-260) that were inadvertently omitted from our original submission. We would once again like to thank the reviewer for generously offering their time and services to critically analyze and provide helpful feedback to improve our manuscript.

Reviewer 3 Report

The authors present a nicely written paper, the experiments are very well designed and a wealth of experimental data is presented.

Solely, the data presentation and interpretation of the ERG data is not sufficient. Indeed, retinal function measured by ERG can be affected by the genetic background of the mice. If this reviewer understands correctly, C57BL/6 were used as wild-type controls, but not Cre-negative dcKO littermates. Therefore, why did the authors not compare the ERG data longitudinally for each genotype between 2 and 5 months and perform an ANOVA analysis comparing the changes between the 2 genotypes over time? Would there then be a significant decrease in retinal function be observed in dcKO mice versus wild-type mice? Please reanalyze the data, and, if appropriate, change the Results and Discussion section accordingly.

Minor comments:

Please add reference of Rhodopsin-Cre mouse line

qPCR: cut-off at what cycle threshold?

Figure Legend 5: not clear to this reviewer what m6, m3 and m2 stands for.

Typos:
Figure 3B and 3D: label correctly dcKO-b-wave

l.304: mitochondrial function

l.362: remove additional period

Author Response

We would like to thank the reviewers for their time and careful review of our article, previously titled: “HK2 and PKM2 are Critical for Photoreceptor Survival and Simultaneous Loss Negates Their Individual Neuroprotective Effects During Retinal Stress”. Please find below our point-by-point rebuttal to address the concerns raised by each reviewer.

Reviewer #3

Besirli and colleagues had previously studied the impact of independently deleting the genes for hexokinase 2 (HK2) and for pyruvate kinase muscle isoform 2 (PKM2) in mouse photoreceptors. In this paper they describe their results with a double photoreceptor knockout of both HK2 and PKM2 in rod photoreceptors in mice.  They report that photoreceptors form normally in the double knockout mice but exhibit a slow retinal degeneration phenotype that progresses faster than the degeneration seen in either HK2 or PKM2 knockouts alone. They show that depletion of these two enzymes results in upregulation of their isozymes HK1 and PKM1. They performed targeted metabolomic analysis to determine that double deletion alters glycolytic metabolism including the depletion of NAD+ but does not appear to alter the accumulation of TCA cycle intermediates or mitochondrial function. They also report some paradoxical results, namely that despite the thinning of the retina and of photoreceptor inner and outer segments, scoptopic and photopic ERG amplitudes are increased in the double knockout mice.  They also found that photoreceptors in the double knockout mice are no more sensitive to experimental retinal detachment than are photoreceptors in wild type mice.  This contrasts with mice deleted only for HK2, which exhibit increased photoreceptor death after retinal detachment.

The paper is clearly written, and the authors describe their method in sufficient detail for others to replicate their results. The experimental results support the authors conclusions, and the Discussion offers plausible explanations for unexpected results.

We would like to thank the reviewer for the careful review of our manuscript.

Specific comments:

  1. Fig. 1A: While band of interest Western Blots are useful for saving space, the authors should provide supplementary figures with the full blot including molecular weight standards to validate the specificity of the reagent antibodies they employed.

We thank the reviewer for this very useful suggestion. We have created a supplemental file to show these data (Supplemental Figures 1 and 3).

  1. Fig. 2: Estimating the thickness of the inner and outer segments by SD-OCT is difficult in mice. The system software permits it, but the result are dicey.These data should be corroborated by representative images using high-resolution light microscopy or electron microscopy. Total retinal thickness might be altered by fluid accumulation, which could be affected by changes in photoreceptor metabolism. Here again, microscopy could corroborate the OCT results (or not).

We thank the reviewer for these insightful comments. We agree that measuring inner segments or outer segments by SD-OCT can be challenging. Indeed, we found that it is difficult to nearly impossible to differentiate between these two layers on OCT when any degeneration or alteration to inner segments/outer segments occur. In order to avoid this issue, we have instead opted to measure the combined inner segment/outer segment layer, which is defined by the external limiting membrane on the inner side, and the RPE layer on the outer side. While not measuring either inner or outer segment thickness individually, this measurement does allow for robust detection of total changes to this layer. For these reasons, we do not report that either inner or outer segments are changed, but that the combined layer thickness is changing. Additionally, multiple groups have published data showing good agreement between OCT and Histological measurements for all layers of the retina, including the inner segment/outer segment region (PMID: 30519502, 19838301, 25360629).

We agree that fluid accumulation in or around photoreceptors or other cells in the retina could be occurring, however this should cause an increase in apparent thickness. Our data show a consistent decrease in thickness as measured by OCT. If any edema was occurring, then it should result in an underestimation of the apparent decline in retinal thickness. For these reasons we believe that histological examination of retinal tissues is better suited for determining changes to gross morphology (total number of cells, rows of nuclei, etc), compared to determining overall tissue thickness.

We agree the ultra-structural examination of inner segments/outer segments via electron microscopy would be useful for better understanding the role of HK2 and PKM2 in proper assembly of these organelles. We believe that these experiments lie outside of the scope of this manuscript and will be added to future manuscripts.

  1. lines 274-290: Since b-wave amplitudes arise from secondary neurons that were not affected by deletion of PKM2 and HK2, the authors should attempt some explanation for this finding in the Discussion. Similarly, since cone photoreceptors were not genetically altered, the authors should attempt to explain the impact on photopic ERG amplitudes.

We thank the reviewer for this pertinent comment. It is interesting that both b-wave and cone dependent ERG amplitudes are affected by this rod-specific knockout. We have added additional text to the discussion (line 519-530) to explain why this may be the case:

“At the same time, the b-wave amplitude was significantly increased despite inner retinal neurons not being affected by the genetic knockout. Similarly, photopic ERG was enhanced, despite cone PRs maintaining normal HK2 and PKM2 expression. Data obtained from human patients with cone-only retinal degeneration (e.g. achromatopsia) have shown defects in rod mediated ERG, despite rods not being affected by the genetic mutation, and is not explained by a decrease in rod photoreceptor numbers [PMID: 25168900]. As it has been shown that changes in cone function can influence rod function, it is plausible that the opposite may also be true. Indeed, patients with rod-driven retinal degeneration (e.g. retinitis pigmentosa) have been shown to display changes to cone function prior to degeneration [PMID: 34795055]. The metabolic microenvironment within the retina, between cones, rods, and RPE cells, is likely altered in these disease processes and may be responsible for the non-cell autonomous changes in function observed.”

  1. Fig. 3: The double knockout b wave is mislabeled as a wave in B and D.

We thank the reviewer for identifying this error. We have updated the figure to correctly label the bars in Fig. 4B&D.

In addition to the changes listed above, we have gone through the entire manuscript to enhance clarity without altering the conclusions of the original manuscript. We have also included an updated description for the retinal detachment and flow cytometric methods (lines 233-260) that were inadvertently omitted from our original submission. We would once again like to thank the reviewer for generously offering their time and services to critically analyze and provide helpful feedback to improve our manuscript.

Reviewer 4 Report

Bersili and colleagues had previously studied the impact of independently deleting the genes for hexokinase 2 (HK2) and for pyruvate kinase muscle isoform 2 (PKM2) in mouse photoreceptors. In this paper they describe their results with a double photoreceptor knockout of both HK2 and PKM2 in rod photoreceptors in mice.  They report that photoreceptors form normally in the double knockout mice but exhibit a slow retinal degeneration phenotype that progresses faster than the degeneration seen in either HK2 or PKM2 knockouts alone. They show that depletion of these two enzymes results in upregulation of their isozymes HK1 and PKM1. They performed targeted metabolomic analysis to determine that double deletion alters glycolytic metabolism including the depletion of NAD+ but does not appear to alter the accumulation of TCA cycle intermediates or mitochondrial function. They also report some paradoxical results, namely that despite the thinning of the retina and of photoreceptor inner and outer segments, scoptopic and photopic ERG amplitudes are increased in the double knockout mice.  They also found that photoreceptors in the double knockout mice are no more sensitive to experimental retinal detachment than are photoreceptors in wild type mice.  This contrasts with mice deleted only for HK2, which exhibit increased photoreceptor death after retinal detachment.

The paper is clearly written, and the authors describe their method in sufficient detail for others to replicate their results. The experimental results support the authors conclusions, and the Discussion offers plausible explanations for unexpected results.

Specific comments:

1. Fig. 1A: While band of interest Western Blots are useful for saving space, the authors should provide supplementary figures with the full blot including molecular weight standards to validate the specificity of the reagent antibodies they employed.

2. Fig. 2: Estimating the thickness of the inner and outer segments by SD-OCT is difficult in mice. The system software permits it, but the result are dicey.  These data should be corroborated by representative images using high-resolution light microscopy or electron microscopy. Total retinal thickness might be altered by fluid accumulation, which could be affected by changes in photoreceptor metabolism. Here again, microscopy could corroborate the OCT results (or not).

3. lines 274-290: Since b-wave amplitudes arise from secondary neurons that were not affected by deletion of PKM2 and HK2, the authors should attempt some explanation for this finding in the Discussion. Similarly, since cone photoreceptors were not genetically altered, the authors should attempt to explain the impact on photopic ERG amplitudes.

4. Fig. 3: The double knockout b wave is mislabeled as a wave in B and D.

Author Response

We would like to thank the reviewers for their time and careful review of our article, previously titled: “HK2 and PKM2 are Critical for Photoreceptor Survival and Simultaneous Loss Negates Their Individual Neuroprotective Effects During Retinal Stress”. Please find below our point-by-point rebuttal to address the concerns raised by each reviewer.

Reviewer #4

The authors present a nicely written paper, the experiments are very well designed and a wealth of experimental data is presented.

We would like to thank the reviewer for their time in critically reading our manuscript and for the positive comments.

Solely, the data presentation and interpretation of the ERG data is not sufficient. Indeed, retinal function measured by ERG can be affected by the genetic background of the mice. If this reviewer understands correctly, C57BL/6 were used as wild-type controls, but not Cre-negative dcKO littermates. Therefore, why did the authors not compare the ERG data longitudinally for each genotype between 2 and 5 months and perform an ANOVA analysis comparing the changes between the 2 genotypes over time? Would there then be a significant decrease in retinal function be observed in dcKO mice versus wild-type mice? Please reanalyze the data, and, if appropriate, change the Results and Discussion section accordingly.

We would like to thank the reviewer for their comment. We apologize for any lack of clarity in describing the generation of the dcKO line. Both the dcKO and WT lines were generated from a double heterozygous in-cross of Hk2wt/fl;Rho-Cre+ x Pkmwt/fl;Rho-Cre+ animals. The resulting generation yielded the expected distribution of genotypes from a double heterozygous in-cross. We selected animals that were double homozygous (i.e. Hk2fl/fl;Pkmfl/fl;Rho-Cre+) and animals that were double wildtype (i.e. Hk2wt/wt;Pkmwt/wt;Rho-Cre+) from the litters produced by double heterozygous in-crosses. We then in-crossed the dcKO animals or the WT animals to produce experimental (Hk2fl/fl;Pkmfl/fl;Rho-Cre+, dcKO) and control (Hk2wt/wt;Pkmwt/wt;Rho-Cre+, WT) animals for all of the data presented in the manuscript. The two lines should be very genetically similar, although not identical. The typical controls used in any conditional knockout experiment would be to maintain expression of the Cre recombinase to rule out any effects of the transgene in relationship to the observed phenotype. We would not expect the presence or absence of the LoxP recombination sites within the genomic DNA to have any effect on our results.

It would have been technically impractical to use true littermate controls for the experiments presented within this manuscript due to the extremely low numbers of animals with the correct genotype produced from a double heterozygous in-cross. Only 1 out of 16 animals produced are expected to be either double homozygous or double wildtype from a double heterozygous cross. At the same time, the Rho-Cre transgene is not homozygous within this line, and therefore less than 100% of the animals produced carry the transgene, further reducing the number of animals which are the correct genotype to between 1 out of every 16 and 32 animals. The WT and dcKO lines we generated should be very genetically similar which will reduce variation due to genetic background as much as practical.

The ERG data presented in this manuscript is not longitudinal, different animals were measured at 2-months and 5-months. As these are independent groups, we believe a student’s t-test is an appropriate measure for determining statistical significance, as has been published previously (PMID: 21031137, 31237438, 27753525). We compared the 2- and 5-month old dcKO and WT groups using a student’s t-test. This analysis showed a significant decrease in both scotopic and photopic amplitudes for WT animals, which has been previously reported (PMID: 11444632). When comparing 2-month to 5-month old dcKO animals we did not find any significant differences between the groups. We have added this analysis to the results, lines 318-326.

Minor comments:

Please add reference of Rhodopsin-Cre mouse line

We apologize for omitting this reference. We have added the original reference for this mouse line (PMID: 16636658) to the methods section, line 101

qPCR: cut-off at what cycle threshold?

We used a cycle threshold cut-off of 35 cycles. We have added this to the methods section, line 160-161.

Figure Legend 5: not clear to this reviewer what m6, m3 and m2 stands for.

We apologize for the lack of clarity on our end. m1 m2, m3, etc. refer to the extra mass conferred by 13C derived from the universally labeled glucose. A m1 molecule would have one 13C carbon, a m2 molecule would have 2 13C carbons incorporated, a m3 molecule would have 3 13C carbons incorporated, etc. The enzymatic breakdown pathways for glucose are well characterized, so the expected pattern of heavy carbons within each molecule is known when starting with a labeled molecule. We have added this explanation to the methods section for added clarity lines 227-232.

Typos: Figure 3B and 3D: label correctly dcKO-b-wave

l.304: mitochondrial function

l.362: remove additional period

We would like to thank the reviewer for identifying these typographical error. We have updated the labels for Figure 3B&D accordingly and corrected the other errors.

In addition to the changes listed above, we have gone through the entire manuscript to enhance clarity without altering the conclusions of the original manuscript. We have also included an updated description for the retinal detachment and flow cytometric methods (lines 233-260) that were inadvertently omitted from our original submission. We would once again like to thank the reviewer for generously offering their time and services to critically analyze and provide helpful feedback to improve our manuscript.

Round 2

Reviewer 1 Report

The authors addressed most of my concerns ans suggestions.

Regarding my first concern about the age of animals, please clarify your answer in the manuscript so it will be clear to the readers.

I accept the manuscript for publication.

Author Response

We would again like to thank the reviewers for making time in their schedules to review our revised manuscript titled “Metabolic alterations caused by simultaneous loss of HK2 and PKM2 leads to photoreceptor dysfunction and degeneration”. Please find below our point-by-point response to the comments from each reviewer.

Reviewer #1:

The authors addressed most of my concerns and suggestions.

We would again like to thank the reviewer for reviewing our revised manuscript.

Regarding my first concern about the age of animals, please clarify your answer in the manuscript so it will be clear to the readers.

We have added our answer to your question to the discussion section, lines 460-469.

I accept the manuscript for publication.

We thank the reviewer for their time and effort in reviewing our manuscript.

We would again like to thank the reviewers for taking time to analyze our revised manuscript. In addition to the changes listed above we have separated our final paragraph into a new conclusion section (line 569).

Reviewer 2 Report

This reviewer raised the concern of not including single knockout PKM2 and HK2 mice in my previous review. In fact, as stated in the rebuttal letter by the authors, these mice have phenotypes. The fact that these mice have phenotypes make them significant for this study. The retinal phenotypes of the single knockout mice are extremely significant in regards to the double knockout PKM2 and HK2 phenotype. I disagree with the authors, and these genotypes should be included in this study.  Inclusion of all the genotypes allows for the comparison of the single knockout phenotypes to the double knockout phenotype, enabling us to determine rates of pathology development, manifestations of pathologies, and responses to experimental conditions such as the experimental retinal detachment. 

Unfortunately, without these controls, many of the conclusions of this manuscript can not be made but rather speculated on.  For example, in the abstract [Lines 17-19], it states 'much more rapidly than either knockout alone' in reference to the double knockout mice having shorter OS segments.  There is no scientific evidence to support this in this paper. 

Author Response

We would again like to thank the reviewers for making time in their schedules to review our revised manuscript titled “Metabolic alterations caused by simultaneous loss of HK2 and PKM2 leads to photoreceptor dysfunction and degeneration”. Please find below our point-by-point response to the comments from each reviewer.

Reviewer #2:

This reviewer raised the concern of not including single knockout PKM2 and HK2 mice in my previous review. In fact, as stated in the rebuttal letter by the authors, these mice have phenotypes. The fact that these mice have phenotypes make them significant for this study. The retinal phenotypes of the single knockout mice are extremely significant in regards to the double knockout PKM2 and HK2 phenotype. I disagree with the authors, and these genotypes should be included in this study.  Inclusion of all the genotypes allows for the comparison of the single knockout phenotypes to the double knockout phenotype, enabling us to determine rates of pathology development, manifestations of pathologies, and responses to experimental conditions such as the experimental retinal detachment.

Unfortunately, without these controls, many of the conclusions of this manuscript can not be made but rather speculated on.  For example, in the abstract [Lines 17-19], it states 'much more rapidly than either knockout alone' in reference to the double knockout mice having shorter OS segments.  There is no scientific evidence to support this in this paper.

We would like to thank the reviewer for their time in reviewing our manuscript. We disagree that it is necessary to include previously published data to draw conclusions on how different phenotypes compare to each other. While we do not replicate previously published results in this manuscript, we do directly describe and reference them to provide insight with how they relate to the phenotype described in this manuscript. It is not necessary to repeat previously published work to compare results between studies. We do in fact show that dcKO animals display earlier degeneration of the inner segments/outer segments by 3 months of age in Figure 2. We cite and report in our discussion that the single HK2 knockout animals do not show significant degeneration until 10 months of age, and PKM2 single knockout animals do not show degeneration until 5 months of age. We additionally compare the phenotype of the dcKO animals to other retinal degeneration models (rd1, rd10). Similarly, it is also not necessary to repeat the phenotypic characterization of these animals to state that the rate of degeneration of dcKO animals is slower.

We would again like to thank the reviewers for taking time to analyze our revised manuscript. In addition to the changes listed above we have separated our final paragraph into a new conclusion section (line 569).